# Direct-Mapping-Based MIMO-FBMC Underwater Acoustic Communication Architecture for Multimedia Signals

**Chin-Feng Lin [1,\*], Tsung-Jen Su [1], Hung-Kai Chang [1], Chun-Kang Lee [1], Shun-Hsyung Chang [2], Ivan A. Parinov [3] and Sergey Shevtsov [4]** 

1   Department of Electrical Engineering, National Taiwan Ocean University, Keelung 20224, Taiwan; healthcarentou@gmail.com (T.-J.S.); smartntou@gmail.com (H.-K.C.); smartntou01@yahoo.com.tw (C.-K.L.)
2   Department of Microelectronics Engineering, National Kaohsuing University of Science and Technology, Kaohsuing 81157, Taiwan; stephenshchang@me.com
3   I. I. Vorovich Mathematics, Mechanics, and Computer Science Institute, Southern Federal University, 344090 Rostov-on-Don, Russia; parinov_ia@mail.ru
4   Head of Aircraft Systems and Technologies Lab at the South Center of Russian Academy of Science, 344006 Rostov-on-Don, Russia; sergnshevtsov@gmail.com
\*   Correspondence: lcf1024@mail.ntou.edu.tw

**Abstract:** In this paper, a direct-mapping (DM)-based multi-input multi-output (MIMO) filter bank multi-carrier (FBMC) underwater acoustic multimedia communication architecture (UAMCA) is proposed. The proposed DM-based MIMO-FBMC UAMCA is rare and non-obvious in the underwater multimedia communication research topic. The following are integrated into the proposed UAMCA: A $2 \times 2$ DM transmission mechanism, a (2000, 1000) low-density parity-check code encoder, a power assignment mechanism, an object-composition petrinet mechanism, adaptive binary phase shift keying modulation and 4-offset quadrature amplitude modulation methods. The multimedia signals include voice, image, and data. The DM transmission mechanism in different spatial hardware devices transmits different multimedia packets. The proposed underwater multimedia transmission power allocation algorithm (UMTPAA) is simple, fast, and easy to implement, and the threshold transmission bit error rates (BERs) and real-time requirements for voice, image, and data signals can be achieved using the proposed UMTPAA. The BERs of the multimedia signals, data symbol error rates of the data signals, power saving ratios of the voice, image and data signals, mean square errors of the voice signals, and peak signal-to-noise ratios of the image signals, for the proposed UAMCA with a perfect channel estimation, and channel estimation errors of 5%, 10%, and 20%, respectively, were explored and demonstrated. Simulation results demonstrate that the proposed $2 \times 2$ DM-based MIMO-FBMC UAMCA is suitable for low power and high speed underwater multimedia sensor networks.

**Keywords:** direct-mapping (DM); multi-input multi-output filter bank multi-carrier (MIMO-FBMC); underwater acoustic multimedia communication architecture (UAMCA)

## 1. Introduction

Underwater acoustic multimedia communication (UAMC) has attracted the interest of many researchers. Zhou et al. [1] conducted a simple investigation of the characteristics of multipath scattering propagation in underwater acoustic channels (UACs), and demonstrated the spatial and frequency correlation between two different propagation paths in UAC. UAC models help fulfill the real-time requirements of transmissions and analyze their bit error rates (BERs) for underwater multimedia signals in UAMC systems. Underwater imaging sensors, and underwater

multimedia sensor networks (UMSNs) can be utilized for underwater explorations and environmental monitoring [2]. Reliable transmission and communication of underwater images are important features to be considered in the design of UMSNs. Sarisaray-Boluk et al. [2] evaluated the performance of underwater image transmission over UMSNs using error concealment and correction algorithms. Their method integrated the forward error correction (FEC) method and adaptive retransmission technologies into an underwater image transmission system. Wang et al. [3] developed an orthogonal frequency division multiplexing (OFDM)-based simulation platform with a channel estimation scheme, and demonstrated that the performance of the UAMC system was heavily dependent on the accuracy of the underwater channel estimation. Woodward et al. [4] proposed an underwater acoustic speech communication system that used a digital signal processor, linear predictive coding, and digital pulse position modulation technologies; and demonstrated that the received speech signals could be recovered clearly. Mahmood et al. [5] developed a software-based underwater digital voice transmission system using the OFDM modulation scheme; the proposed technology can be used with mobile underwater vehicles at speeds of up to 30 nautical miles per hour. A real-time evaluation model of underwater voice communication, and a watermark algorithm with reference voice feature information are presented in [6]. Multiple-input multiple-output (MIMO) schemes with spatial multiplexing gain, filter bank multi-carrier (FBMC) modulation, the offset quadrature amplitude modulation (OQAM), and preamble-based channel estimation are integrated into a proposed underwater acoustic video transmission system, and the BER in the time and frequency dispersions in the UAC are discussed in [7].

The FBMC transmission method outperforms OFDM in terms of spectral efficiency gains [8]; however, the interference caused by the transmitted data in the time-frequency domain of a MIMO-FBMC system is a challenge, and a fast Fourier transform (FFT)-based FBMC transmission strategy is proposed to achieve low interference. Caus et al. [9] presented the MIMO precoding and decoding matrices using an OQAM/FBMC system with a multi-stream transmission strategy in high coherence bandwidth MIMO channels. Wang [10] adopted a sparse channel estimation method to realize accurate reconstruction of the channel information by adaptively selecting the support set, leading to a decrease in the intrinsic interference in a MIMO-FBMC/OQAM system. Higher spectral efficiency gains can be achieved using MIMO-FBMC/OQAM in comparison with those achieved using FBMC/OQAM [11]. Wang et al. [11] utilized an extended preamble structure channel estimation method with a symmetrical pattern to decrease the inherent imaginary interference in the MIMO-FBMC/OQAM technology, thereby achieving lower peak to average power ratios (PAPRs), BERs, and mean square errors (MSEs), in comparison with those obtained using conventional MIMO-FBMC/OQAM channel estimation methods. In [12], the asymptotic performance of an FBMC-based system using massive MIMO technology was studied, and it was observed that the inter-symbol and inter-carrier interferences did not decrease with an increase in the number of base station antennas. Additionally, it was found that an efficient equalization method could decrease the channel distortions caused by inter-symbol and inter-carrier interferences. Zafar et al. [13] analyzed the performance of the MIMO-FBMC system theoretically, and integrated the filter output truncation method into the analysis model. The proposed method increases the signal to interference ratio of each symbol, resulting in increased BERs; overhead and filter truncations were not required in the proposed method. Sim et al. [14] demonstrated a two-tap MIMO maximum likelihood detection scheme using a two-dimensional ordering algorithm and used it to reduce multi-dimensional residual interference in the MIMO-FBMC system. The method enabled efficient cancellation of interferences in the system.

Woodward et al. [15] proposed a Filtered Multitone (FMT) modulation underwater acoustic communications system, integrated with low-complexity channel-estimation-based minimum MSEs adaptive turbo equalization, (7,5) convolutional channel coding, adaptive binary phase shift keying (BPSK), and eight phase shift keying modulation. The BERs and data symbol error (DSE) rates of the proposed system are explored. Qiao et al. [16] proposed a 2 × 2 MIMO OFDM-based high speed underwater acoustic transmission system, with low density parity check (LDPC) channel coding,

orthogonal matching pursuit (OMP) channel estimation, and minimum mean-squared error (MMSE) equalization methods integrated into the underwater acoustic transceiver. The BERs of the transceiver were studied. Amini et al. [17] adopted the FBMC technology with a cost function to optimize the filterbank prototype filter in doubly dispersive underwater acoustic communication channels. Direct-mapping (DM) orthogonal variable spreading factor (OVSF)-based [18], MIMO-OFDM-based [19], Gold sequence (GS) MIMO OVSF/OFDM-based [20], and OVSF/OFDM-based [21] underwater acoustic transmission methods have also been studied. The DM-based MIMO-FBMC underwater acoustic transmission method was used to transmit audio [22] and data [23] signals. In this paper, a DM-based MIMO-FBMC underwater acoustic multimedia communication architecture (UAMCA) is presented, as are its applications in the transmission of multimedia signals. The DM-based MIMO-FBMC UAMCA method is proposed in Section 2 of this paper, while the simulation results are presented in Section 3, and the concluding remarks are presented in Section 4.

## 2. System Models

Figure 1 draws the proposed 2 × 2 DM-based MIMO-FBMC UAMCA. The multimedia signals include voice, image and data signals. The UAMCA includes the following features: A 2 × 2 DM-based MIMO transmission strategy; adaptive BPSK or 4-OQAM modulations; wireless multimedia packet-by-packet transmission method; a power assignment algorithm; (2000, 1000) LDPC code encoder with a code rate of 1/2, column weight of 3, and row weight of 6; an object-composition petrinet (OCPN) mechanism [24]; and serial-to-parallel and parallel-to-serial mechanisms. The PHYDYAS FBMC modulation architecture [25] with a 64-point inverse fast Fourier transform (IFFT) and two polyphase networks (PPNs) are integrated into the UAMCA, and the design parameters of the proposed 2 × 2 DM-based MIMO-FBMC UAMCA are listed in Table 1.

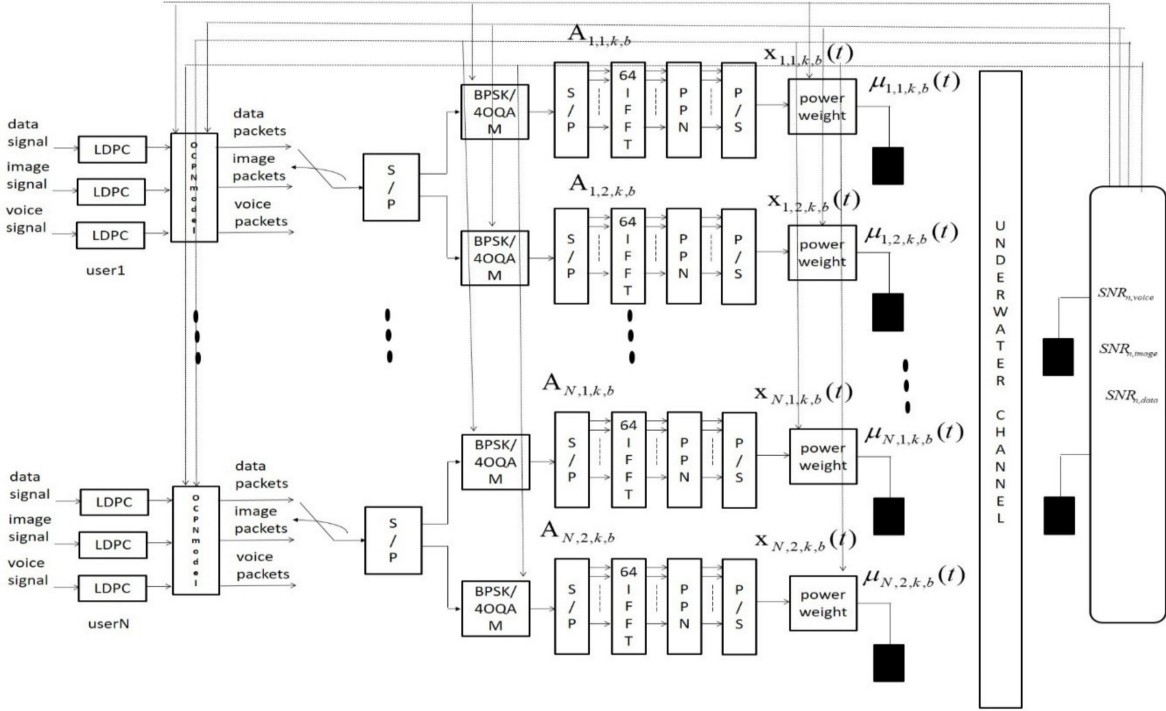

**Figure 1.** The proposed 2 × 2 DM-based (direct-mapping) MIMO-FBMC UAMCA (multi-input multi-output)-(filter bank multi-carrier) (underwater acoustic multimedia communication architecture).

**Table 1.** The parameters of the proposed $2 \times 2$ DM-based MIMO-FBMC UAMCA.

| Technology | Technology Characteristics |
|---|---|
| FBMC modulation | PHYDAS FBMC technology [25] |
| FFT size | 64-points |
| Channel coding | (2000, 1000) LDPC code encoder with a code rate of 1/2, a column weight of 3, a row weight of 6 |
| MIMO | $2 \times 2$ DM |
| Power levels | $0.09, 0.18, 0.27, \ldots, 1$ |
| BER limits for voice transmission | $10^{-3}$ |
| BER limits for image transmission | $10^{-4}$ |
| BER limits for data transmission | $10^{-5}$ |

Voice, image, and data multimedia signa ls were the inputs to the (2000, 1000) LDPC encoders, and voice, image, and data multimedia LDPC coding bit streams were extracted as output. The multimedia real-time transmission was calculated using the OCPN mechanism, and as a result, the outputs were then used as the inputs to the OCPN mechanism, which yielded voice, image, and data multimedia LDPC coding packets as output. The voice, image, and data multimedia LDPC coding packets were input into the $2 \times 2$ DM-based MIMO mechanism using a serial-to-parallel method, and the $2 \times 2$ DM-based MIMO multimedia packets were extracted as outputs. Different voice, image, and data multimedia LDPC coding packets use different spatial hardware to achieve high-speed underwater multimedia communication in the DM-based transmission mechanism. The $2 \times 2$ DM-based MIMO multimedia packets were input into adaptive BPSK or 4-OQAM modulations, which adaptively yielded $2 \times 2$ DM-based MIMO multimedia modulated packets. The adaptive modulation yielded $2 \times 2$ DM-based MIMO multimedia modulated packets that were in the serial-to-parallel method, 64-point IFFT, two PPNs, and parallel-to-serial method; and $2 \times 2$ DM-based MIMO-FBMC multimedia modulated packets with power assignment schemes were extracted as final outputs. In the proposed $2 \times 2$ DM-based MIMO-FBMC UAMCA, the BER limit values for voice, image, and data packets are assumed to be equal to or below $10^{-3}$, $10^{-4}$, and $10^{-5}$, respectively. This assumes that the $2 \times 2$ DM-based MIMO-FBMC UAMCA, has $N$ users, $M$ FBMC symbols in a DM-based MIMO-FBMC transmission packet, and $K$ sub-carriers in an FBMC symbol. There are $P$ transmission underwater microphones, and $Q$ receiver hydrophones in the UAMCA. The baseband $2 \times 2$ DM-based MIMO-FBMC adaptive transmitted signal is given as follows:

$$x_{n,p,k,b}(t) = \sum_{b=1}^{MK} \sum_{k=1}^{K-1} A_{n,p,k,b} \phi_{p,k,b}(t) \tag{1}$$

where $A_{n,p,k,b}$ is a BPSK or 4-OQAM symbol with a symbol period $T$ for the $n$th user, transmitted on the $p$th underwater microphones and the $k$th subcarrier at the instant $bT$. $p$ is one or two. Assuming $\phi_{p,k,b}(t)$ is the time-frequency component of *pfir(t)* for BPSK modulation, where *pfir(t)* is the prototype filter impulse response, $\phi_{p,k,b}(t)$ can be described as follows:

$$\phi_{p,k,b}(t) = pfir(t)e^{j\frac{2\pi}{T}M} \tag{2}$$

If $\phi_{p,k,b}(t)$ is the time-frequency component of *pfir(t)* for 4-OQAM modulation, $\phi_{p,k,b}(t)$ can be described as follows:

$$\phi_{p,k,b}(t) = pfir(t - \frac{bT}{2})e^{j\frac{2\pi}{T}M}e^{j\psi_{p,k,b}} \tag{3}$$

where $\psi_{p,k,b}$ is the phase component of the $p$th transmission underwater microphones, described as $\frac{\pi}{2}(k+b) - \pi kb$.

As a result, the adaptive $2 \times 2$ DM-based MIMO-FBMC signal with the power assignment method can be represented as follows:

$$\mu_{n,p,k,b}(t) = \sum_{b=1}^{MK} \sum_{k=1}^{K-1} r_{n,p} A_{n,p,k,b} \phi_{p,k,b}(t) \tag{4}$$

where $r_{n,p}$ is the transmission power weighting of the *nth* user, transmitted on the *p*th underwater microphones. The signal received on the *q*th hydrophone, for the *nth* user, is expressed as follows:

$$y_{n,q,k,b}(t) = \sum_{p=1}^{P} \mu_{n,p,k,b}(t) * h_{p,q}(t) + w_{p,q}(t) \tag{5}$$

where $h_{p,q}(t)$ is the underwater multipath tapped-delay line channel impulse response of the *p*th transmission underwater microphone and the *q*th receiver hydrophone, for the *nth* user. *p* is 1 or 2, and *q* is 1 or 2. The SNRs (signal-to-noise) of the *n*-th user, for received voice, image, and data multimedia packets, can be detected using the double window detection algorithm [26] and are denoted with the symbols '$SNR_{n,voice}$', '$SNR_{n,image}$', and '$SNR_{n,data}$', respectively. The proposed underwater multimedia transmission power assignment algorithm (UMTPAA) for the $2 \times 2$ DM-based MIMO-FBMC UAMCA technology is summarized as follows:

Step 1: With the output information obtained from the OCPN mechanism, assign the transmission packet rates for underwater voice, image, and data signals

Step 2: Adopt the appropriate modulation scheme to meet the quality of services requirements for underwater voice, image, and data multimedia communication.

Step 3: Set up the initial underwater transmission power level of $r_{n,p}$ as 0.54 for voice, image, and data multimedia signals.

Step 4: Measure the received $SNR_{n,voice}$, $SNR_{n,image}$, and $SNR_{n,data}$ of the *n*-th user, for received underwater voice, image, and data multimedia packets, respectively.

Step 5: If the measured $SNR_{n,voice}$, $SNR_{n,image}$, and $SNR_{n,data}$ of the received underwater voice, image, and data multimedia packets, respectively, exceed the underwater multimedia threshold SNRs at the required multimedia BERs of $10^{-3}$, $10^{-4}$, and $10^{-5}$, respectively, for voice; and the image or data packets are achieved, then update $r_{n,p}$ as $r_{n,p} = r_{n,p} - \Delta$, where $\Delta$ is equal to 0.09.

　　If $r_{n,p} \geq 0.09$, go to Step 4; otherwise, go to Step 7.

Step 6: If the measured $SNR_{n,voice}$, $SNR_{n,image}$, and $SNR_{n,data}$ of the received underwater voice, image, and data multimedia packets, respectively, is less than the underwater multimedia threshold SNRs at the required multimedia BERs of $10^{-3}$, $10^{-4}$, and $10^{-5}$, respectively, for voice; and the image or data packets is achieved, then update $r_{n,p}$ as $r_{n,p} = r_{n,p} + \Delta$, where $\Delta$ is equal to 0.09.

　　If $r_{n,p} \leq 1$, go to Step 4; otherwise, go to Step 7.

Step 7: Downgrade the modulation scheme. If the modulation scheme is not 4-OQAM, then go to Step 4.

Step 8: Upgrade the modulation scheme. If the modulation scheme is not BPSK, go to Step 4.

The receiver structure has the inverse function of the transmitter structure. The $2 \times 2$ DM-based MIMO-FBMC received multimedia packets were demodulated using a $2 \times 2$ DM-based MIMO-FBMC demodulator, and the $2 \times 2$ DM-based MIMO-FBMC multimedia demodulation packets were extracted as output. The $2 \times 2$ DM-based MIMO-FBMC multimedia demodulation packets were the inputs to the adaptive BPSK/4-OQAM demodulator, and the adaptive BPSK/4-OQAM multimedia demodulation packets were extracted as output. The adaptive BPSK/4-OQAM multimedia demodulation packets

were the inputs to (2000, 1000) LDPC decoders, and the voice, image, and data multimedia LDPC decoding signals were extracted as output.

## 3. Simulation Results

The MIMO underwater channel model in [27] was adopted, which exhibited a carrier central frequency of 11.5 kHz, an underwater channel bandwidth of 4 kHz, and a transmission distance of 1 km. Figure 2 illustrates the BER performances of the 2 × 2 DM-based MIMO-FBMC UAMCA with channel estimation errors (CEEs) of 0% (perfect channel estimation), 5%, 10%, and 20%. In Figures 2–5, the symbols 'o', and 'Δ' denote the BPSK and 4-OQAM modulation technology, respectively. In Figures 2–5, the colors 'black', 'red', 'blue', and 'fuchsia' denote the proposed UAMCA with CEEs of 0%, 5%, 10%, and 20%, respectively. The BER performances of the proposed UAMCA with CEEs of 0%, 5%, 10%, and 20%, using BPSK modulation technology, and for a signal-to-noise ratio (SNR) of 12.59 dB, are $2 \times 10^{-6}$, $2.67 \times 10^{-6}$, $4.67 \times 10^{-6}$, and $2.13 \times 10^{-5}$, respectively. The BER performances of the proposed UAMCA using 4-OQAM modulation with all other conditions retained from the BPSK system, are $2.93 \times 10^{-5}$, $5.73 \times 10^{-5}$, 0.00011, and 0.00041, respectively. The simulation results demonstrate that the BER performances of the proposed UAMCA with BPSK modulation technology are better than those of the system with 4-OQAM modulation technology. Furthermore, the BER performances of the proposed UAMCA degraded as the CEEs increased. The transmission BERs requirements for underwater voice, image, and data signals are $10^{-3}$, $10^{-4}$, and $10^{-5}$, respectively. Consequently, low transmission power can be achieved for underwater voice, image, and data multimedia communication. The power-saving (PS) ratio of the DM-based MIMO-FBMC UAMCA is defined as follows:

$$PS = (1 - r_{n,p}) \times 100\% \qquad (6)$$

Figures 3–5, depict the PS ratios of the DM-based MIMO-FBMC UAMCA with BERs of $10^{-3}$, $10^{-4}$, and $10^{-5}$, respectively, for CEEs of 5%, 10%, and 20%. Figure 3 depicts that the $N_o$ values for the proposed UAMCA using BPSK modulation technology, with the BER of $10^{-3}$ and a PS ratio of 27%, are 0.1238, 0.1105, and 0.0869, respectively. $N_o$ is the power spectral density of additive white Gaussion noise (AWGN).

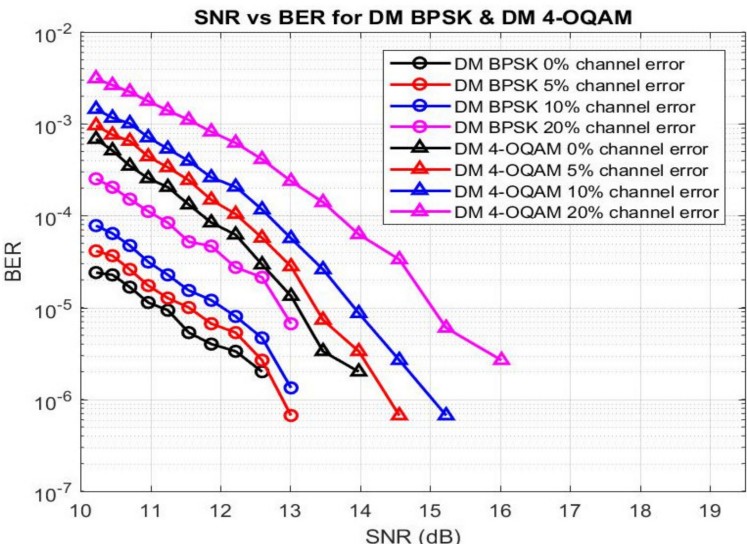

**Figure 2.** The BER performances of the 2 × 2 DM-based MIMO-FBMC UAMCA with CEEs of 0%, 5%, 10%, and 20%.

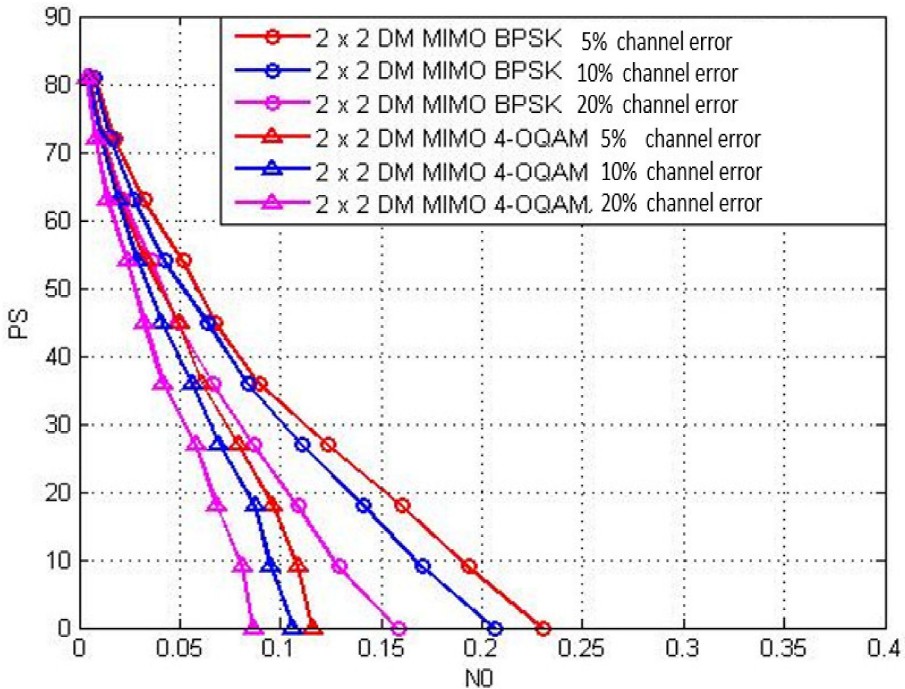

**Figure 3.** The PS ratio of the $2 \times 2$ DM-based MIMO-FBMC UAMCA with the BER of $10^{-3}$, for CEEs of 5%, 10%, and 20%.

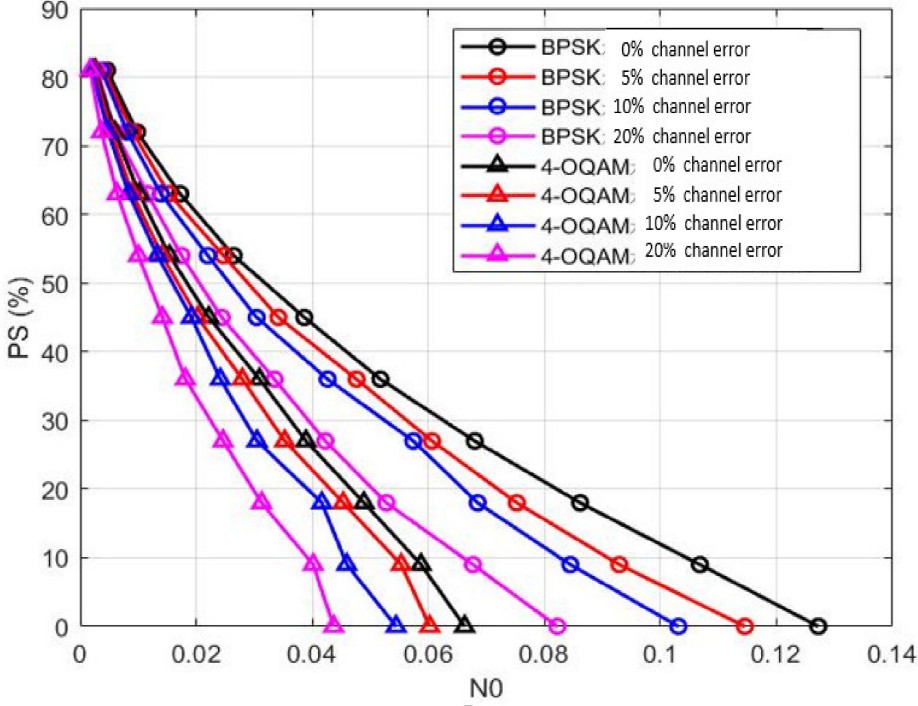

**Figure 4.** The PS ratio of the $2 \times 2$ DM-based MIMO-FBMC UAMCA with the BER of $10^{-4}$, for CEEs of 5%, 10%, and 20%.

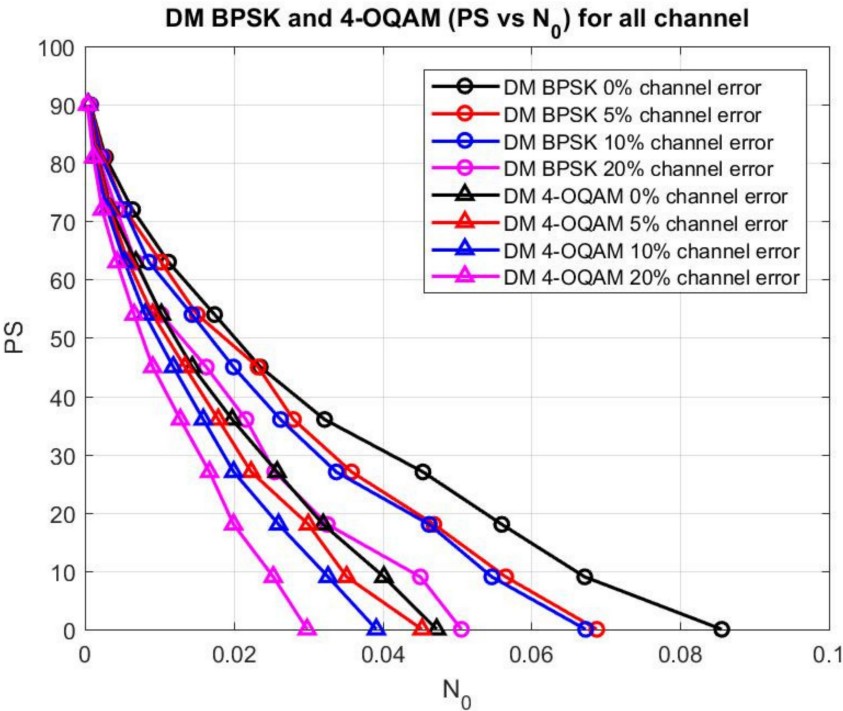

**Figure 5.** The PS ratio of the DM-based MIMO-FBMC UAMCA with the BER of $10^{-5}$, for CEEs of 5%, 10%, and 20%.

The $N_o$ values for the proposed UAMCA using 4-OQAM modulation technology, with the BER of $10^{-3}$ and a PS ratio of 27%, are 0.0791, 0.0695, and 0.0581, respectively. Figure 4 depicts that the $N_o$ values for the proposed UAMCA using BPSK modulation technology, with the BER of $10^{-4}$ and a PS ratio of 27%, are 0.0606, 0.0574, and 0.0422, respectively. The $N_o$ values for the proposed UAMCA using 4-OQAM modulation technology, with the BER of $10^{-4}$ and a PS ratio of 27%, are 0.0352, 0.0304, and 0.0246, respectively. Figure 5 depicts that the $N_o$ values for the proposed UAMCA using BPSK modulation technology, with the BER of $10^{-5}$ and a PS ratio of 27%, are 0.0358, 0.0337, and 0.0254, respectively. Finally, The $N_o$ values for the proposed UAMCA using 4-OQAM modulation technology, with the BER of $10^{-5}$ and a PS ratio of 27%, are 0.0223, 0.0199, and 0.0167, respectively. With increasing transmission BER requirements for underwater voice, image, and data signals, $N_o$ decreases, with the same PS ratio. As the CEEs increase, $N_o$ decreases, with the same PS ratio. The $N_o$ of 4-OQAM is less than that of BPSK, with the same PS ratio. Consequently, the PS ratios of BPSK are greater than those of 4-OQAM under BERs of $10^{-3}$, $10^{-4}$, and $10^{-5}$. As the CEEs decrease, the PS ratio increases. Figure 6 illustrates the original (black), and received (red) voice signals using the $2 \times 2$ DM-based MIMO-OFDM UAMCA with a BER of $10^{-2}$ and a CEE of 10%, and with BPSK modulation technology. The mean square error (MSE) performance is $2.68 \times 10^4$, and the received voice is unclear. MSE is the evaluation parameter used for the received voice and the MSE of the original and received voice signals using the $2 \times 2$ DM-based MIMO-OFDM UAMCA is given as follows:

$$MSE = \frac{1}{W} \sum_{i-1}^{W} \left( V_i - \hat{V}_i \right)^2 \tag{7}$$

where $V_i$ is the original voice signal in the underwater simulation transmission experiment (USTE), $\hat{V}_i$ is the received voice signal in the USTE, and $W$ is the length of the voice signal. Figure 7 illustrates the original (black), and the received (red) voice signals using the $2 \times 2$ DM-based MIMO-FBMC UAMCA using BPSK modulation technology with a BER of $10^{-3}$ and a CEE of 10%. The MSE performance is $6.90 \times 10^{-5}$, and the received voice is clear to human listeners. Figure 8 illustrates

the original image signal in the USTE. Figure 9 illustrates the received image signal using the 2 × 2 DM-based MIMO-FBMC UAMCA using BPSK modulation technology with a BER of $10^{-3}$ and a CEE of 10%. The received image evaluation parameter is peak SNR (PSNR). The PSNR parameter in the 2 × 2 DM-based MIMO-FBMC UAMCA is defined as follows:

$$IMSE = \frac{1}{st} \sum_{i=0}^{s-1} \sum_{j=0}^{t-1} [A(i,j) - B(i,j)]^2 \tag{8}$$

$$PSNR = 10 \log_{10} \left( \frac{Max(A(i,j))^2}{IMSE} \right) \tag{9}$$

where *IMSE* is the image MSE of an image signal comprised of $s \times t$ pixels. *A(i,j)* and *B(i,j)* are the matrices containing the pixel values, for the original and received image signals, respectively. The PSNR of the received image is 53.50 dB in Figure 9, and the received image has a snowflake effect.

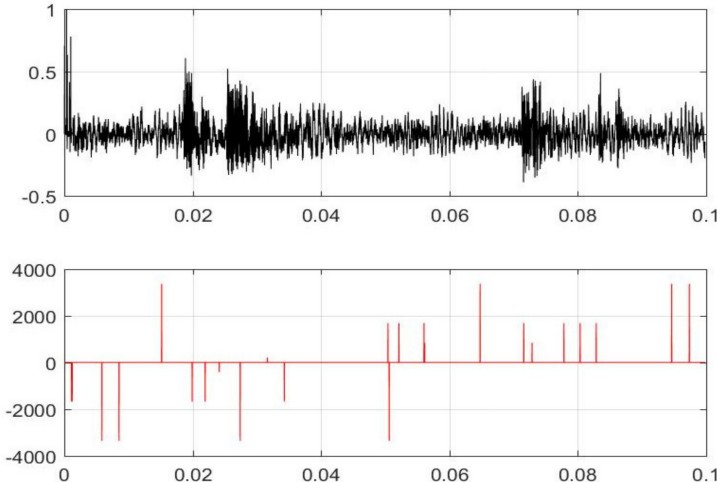

**Figure 6.** The original (black), and the received (red) voice signals using the 2 × 2 DM-based MIMO-FBMC UAMCA using BPSK modulation technology with a BER of $10^{-2}$ and a CEE of 10%.

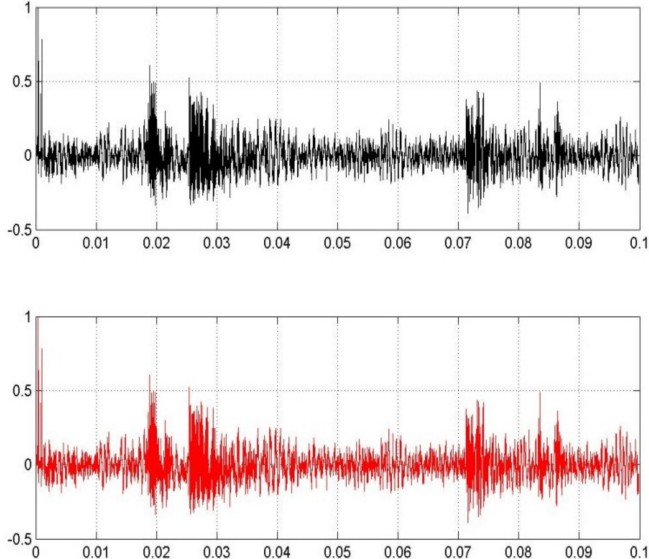

**Figure 7.** The original (black), and the received (red) voice signals using the 2 × 2 DM-based MIMO-FBMC UAMCA using BPSK modulation technology with a BER of $10^{-3}$ and a CEE of 10%.

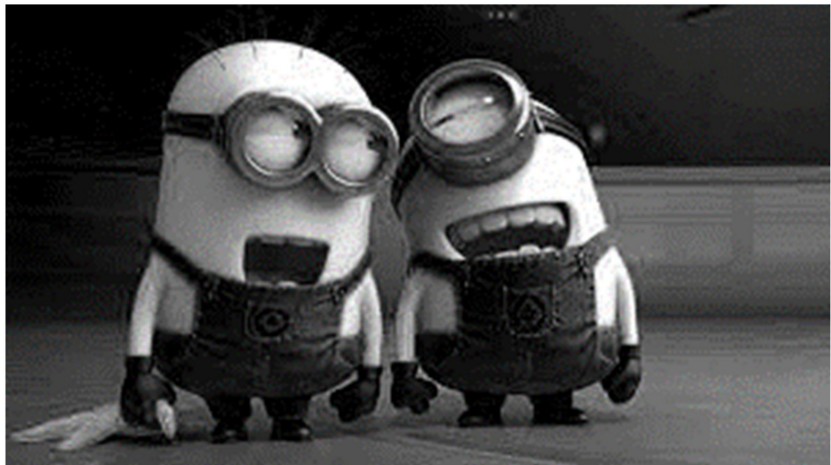

**Figure 8.** The original image signal in the USTE.

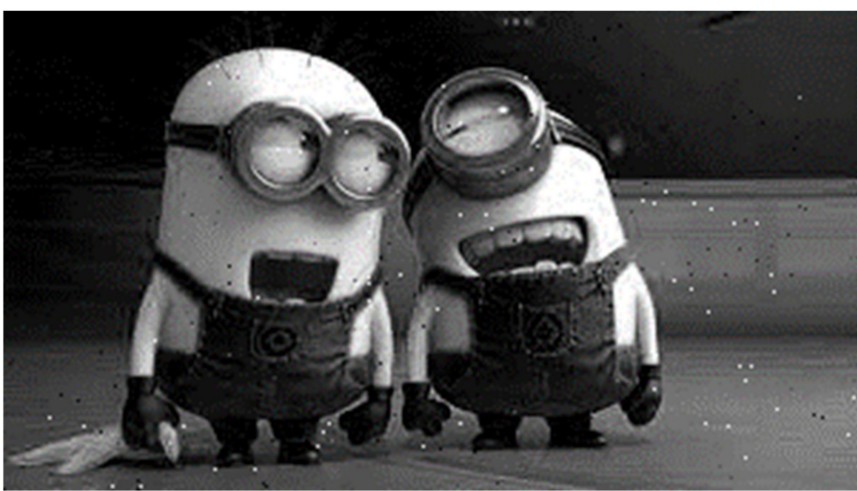

**Figure 9.** The received image signal using the 2 × 2 DM-based MIMO-FBMC UAMCA using BPSK modulation technology with a BER of $10^{-3}$ and a CEE of 10%.

Figure 10 illustrates the received image signal using the 2 × 2 DM-based MIMO-FBMC UAMCA using BPSK modulation technology with a BER of $10^{-4}$ and a CEE of 10%. The PSNR of the received image is 67.52 dB in Figure 10, and the received image is clear to human viewers. Figure 11 shows the data symbol error (DSE) rates of the 2 × 2 DM-based MIMO-FBMC UAMCA with a PCE, CEEs of 5%, 10%, and 20%, with BPSK and 4-OQAM modulation technologies. The 100,000 data symbols were simulated in the underwater data transmission experiment. The DSE rates for the proposed UAMCA with CEEs of 0%, 5%, 10%, and 20%, respectively, and BPSK modulation technologies, are 0.00006, 0.00008, 0.00014, and 0.00064, with the SNR of 12.59 dB. The DSE rates of the proposed UAMCA with 4-OQAM modulation technologies, are 0.00088, 0.00168, 0.00346, and 0.01222, under the same conditions. The DSE rate performance for the proposed UAMCA with BPSK modulation technology is better than that with 4-OQAM modulation technology, and as the CEEs increase, the DSE rate performance decreases.

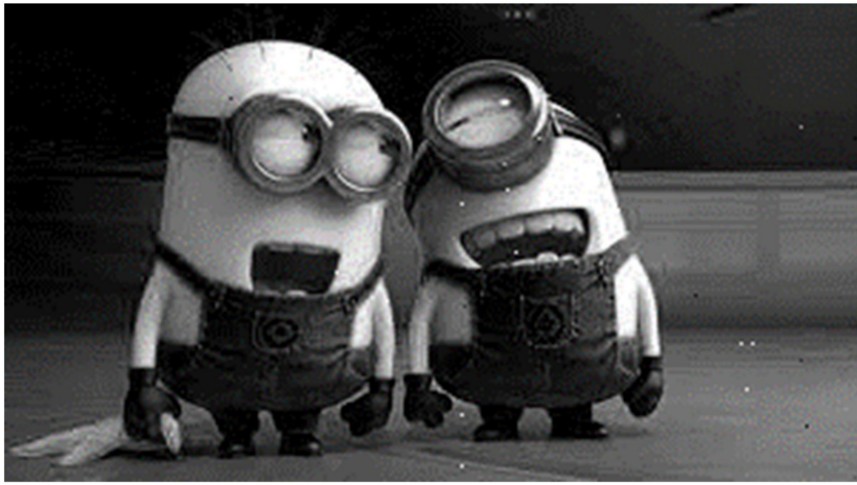

**Figure 10.** The received image signal using the 2 × 2 DM-based MIMO-FBMC UAMCA using BPSK modulation technology with a BER of $10^{-4}$ and a CEE of 10%.

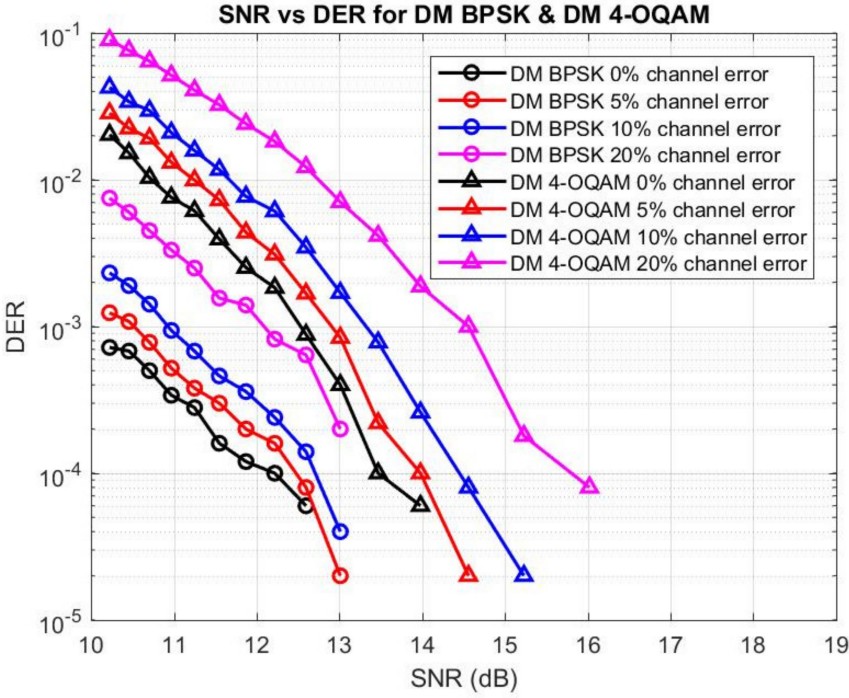

**Figure 11.** The data symbol error (DSE) rates of the 2 × 2 DM-based MIMO-FBMC UAMCA with a PCE, and CEEs of 5%, 10%, and 20%, with BPSK and 4-OQAM modulation technologies.

## 4. Conclusions

The 2 × 2 DM-based MIMO-FBMC UAMCA was described herein and the MIMO underwater channel model with a transmission distance of 1 km was used to simulate its performance. The performances of BERs and PS ratios for the proposed UAMCA with BPSK and 4-OQAM modulation technologies were demonstrated. The UAMCA evaluation parameters, MSEs, PSNRs, and DSE rates, respectively, for underwater voice, image, and data signal transmission were explored. The quality of the received voice signals with BERs of $10^{-2}$, and $10^{-3}$, were presented, and the quality of the received image signals with BERs of $10^{-3}$, and $10^{-4}$, were demonstrated. Simulation results demonstrated that the BER performances of the proposed UAMCA with BPSK modulation technology were better than those of the proposed UAMCA with 4-OQAM modulation technology. Furthermore, the BER

performances of the proposed UAMCA degraded as the CEEs increased. As the CEEs decreased, the PS ratios increased.

The DSE rates performance of the proposed UAMCA with BPSK modulation technology were better than those of the proposed UAMCA with 4-OQAM modulation technology. As the CEEs increased, the DSE rates performance decreased. Thus, the proposed $2 \times 2$ DM-based MIMO-FBMC UAMCA is a feasible underwater low power multimedia communication solution, suitable for underwater low power and high speed underwater multimedia sensor networks. In the future, a space time block code (STBC)-based MIMO-FBMC UAMCA will be explored to achieve low transmission BERs.

**Author Contributions:** Conceptualization, C.-F.L. and S.-H.C.; methodology, C.-F.L., and S.-H.C.; software, C.-F.L., T.-J.S., H.-K.C., C.-K.L.; formal analysis, C.-F.L., and S.-H.C.; investigation, C.-F.L., and S.-H.C.; writing—original draft preparation, C.-F.L., and S.-H.C.; writing—review and editing, C.-F.L., S.-H.C., I.A.P., and S.S. All authors have read and agreed to the published version of the manuscript.

**Funding:** This research was funded by the grant from The Ministry of Science and Technology of Taiwan, under contract No. MOST 107-2221-E-992-027, and MOST 105-2923-E-022-001-MY3.

**Conflicts of Interest:** The authors declare no conflict of interest.

## Abbreviations

| | |
|---|---|
| AWGN | additive white Gaussion noise |
| BER | bit error rate |
| BPSK | binary phase shift keying |
| CEEs | channel estimation errors |
| DM | direct mapping |
| DSE | data symbol error |
| FBMC | filter bank multi-carrier |
| FEC | forward error correction |
| FMT | Filtered Multitone |
| FFT | fast Fourier transform |
| GS | Gold sequence |
| IFFT | inverse fast Fourier transform |
| LDPC | low-density parity-check code |
| MIMO | multi-input multi-output |
| MMSE | minimum mean-squared error |
| MSE | mean square error |
| OCPN | object-composition petrinet |
| OFDM | orthogonal frequency division multiplexing |
| OMP | orthogonal matching pursuit |
| OQAM | offset quadrature amplitude modulation |
| OVSF | orthogonal variable spreading factor |
| PAPR | peak to average power ratio |
| PPNs | polyphase networks |
| PS | power-saving |
| PSNR | peak SNR |
| SNR | signal-to-noise ratio |
| STBC | space time block code |
| UAC | underwater acoustic channel |
| UAMC | underwater acoustic multimedia communication |
| UAMCA | underwater acoustic multimedia communication architecture |
| UMSNs | underwater multimedia sensor networks |
| UMTPAA | underwater multimedia transmission power allocation algorithm |
| USTE | underwater simulation transmission experiment |

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
