# Peer review of "Direct-Mapping-Based MIMO-FBMC Underwater Acoustic Communication Architecture for Multimedia Signals"

_applsci, doi:10.3390/app10010233_

Round 1

Reviewer 1 Report

The paper presents a Multi-input multi-output (MIMO) filter bank multi-carrier (FBMC) underwater acoustic multimedia transmission architecture. The authors should address the following issues:

The contribution of the paper should be written in detail. In particular, how the posed work is different from the existing work should be discussed clearly. Figure 12 and 13 can be made more readable. The future work can be added in the conclusion.

Author Response

The authors acknowledge the valuable comments of the reviewers.

Reviewer 1:

The paper presents a Multi-input multi-output (MIMO) filter bank multi-carrier (FBMC) underwater acoustic multimedia transmission architecture. The authors should address the following issues:

The contribution of the paper should be written in detail. In particular, how the posed work is different from the existing work should be discussed clearly.

In the paper, a direct-mapping (DM)-based Multi-input multi-output (MIMO) filter bank multi-carrier (FBMC) underwater acoustic multimedia communication architecture (UAMCA) is proposed. The proposed DM-based MIMO-FBMC UAMCA is rare and non-obvious in the underwater multimedia communication research topic. The following are integrated into the proposed UAMCA: a 2 x 2 DM transmission mechanism, a (2000, 1000) low-density parity-check code encoder, a power assignment mechanism, an object-composition petrinet mechanism, adaptive binary phase shift keying modulation and 4-offset quadrature amplitude modulation methods. The multimedia signals include voice, image, and data. The DM transmission mechanism in different spatial hardware devices transmits different multimedia packets. The proposed underwater multimedia transmission power allocation algorithm (UMTPAA) is simple, fast, and easy to implement, and the threshold transmission bit error rates and real-time requirements for voice, image, and data signals can be achieved using the proposed UMTPAA. The BERs of the multimedia signals, data symbol error rates of the data signals, power saving  ratios of the voice, image, and data signals, mean square errors of the voice signals, and peak signal-to-noise ratios of the image signals, for the proposed UAMCA with a perfect channel estimation, and channel estimation errors of 5%, 10%, and 20%, respectively, were explored and demonstrated. Simulation results demonstrate that the proposed 2 x 2 DM-based MIMO-FBMC UAMCA is suitable for low power and high speed underwater multimedia sensor networks.

Figure 12 and 13 can be made more readable.

Figures 12 and 13 have been deleted.

The future work can be added in the conclusion. 

In the future, a space time block code (STBC)-based MIMO-FBMC UAMCA will be explored to achieve low transmission BERs.

Reviewer 2 Report

This paper deals with a Multi-input multi-output (MIMO) filter bank multi-carrier (FBMC) underwater acoustic multimedia transmission architecture (UAMTA). The proposed UAMTA considers 2 x 2 direct mapping (DM) transmission mechanism, (2000, 1000) low-density parity-check code (LDPC) encoder, a power assignment mechanism, an object-composition petri net (OCPN) mechanism, adaptively binary phase shift keying (BPSK) modulation and 4-offset quadrature amplitude modulation (OQAM) methods. Simulation results show that the proposed 2 x 2 DM-based MIMO-FBMC transmission architecture is suitable for underwater multimedia communication.

After my own reading, I have the following comments before this paper is accepted, and major revision is recommended.

[1] The author should rewrite the abstract as an overall summary of the paper. In particular, the author needs to add more to the underwater multimedia transmission power allocation algorithm proposed in this paper.

[2] SNIR is mentioned in Figure 1 of this paper. The author needs to write the full name of the SNIR. And there is only a transmitter structure, no receiver structure. I wonder what the receiver structure is.

[3] There is ‘2 UAMTA’ in line 149. I wonder what it means.

[4] There is a sentence in line 197 that the MIMO-FBMC channel estimation is assumed. I don't know what it means. Does it mean complete of channel estimation process?

[5] This paper considers 2x2 DM transmission mechanism, (2000, 1000) LDPC encoder, a power assignment mechanism, an OCPN mechanism, adaptively BPSK modulation and 4-OQAM methods. I think it is necessary to explain clearly what the new proposal is.

[6] From the simulation results, the advantages of the proposed algorithm are not clear. If the channel estimation error rate is high, it is not surprising that the performance of the communication system is poor. Simulation results should be added to clearly show the advantages of the proposed technique.

Author Response

The authors acknowledge the valuable comments of the reviewers.

Reviewer 2:

This paper deals with a Multi-input multi-output (MIMO) filter bank multi-carrier (FBMC) underwater acoustic multimedia transmission architecture (UAMTA). The proposed UAMTA considers 2 x 2 direct mapping (DM) transmission mechanism, (2000, 1000) low-density parity-check code (LDPC) encoder, a power assignment mechanism, an object-composition petri net (OCPN) mechanism, adaptively binary phase shift keying (BPSK) modulation and 4-offset quadrature amplitude modulation (OQAM) methods. Simulation results show that the proposed 2 x 2 DM-based MIMO-FBMC transmission architecture is suitable for underwater multimedia communication.

After my own reading, I have the following comments before this paper is accepted, and major revision is recommended.

[1] The author should rewrite the abstract as an overall summary of the paper. In particular, the author needs to add more to the underwater multimedia transmission power allocation algorithm proposed in this paper.

The abstract has been rewritten.

Please refer the abstract section.

[2] SNIR is mentioned in Figure 1 of this paper. The author needs to write the full name of the SNIR. And there is only a transmitter structure, no receiver structure. I wonder what the receiver structure is.

SNIR has been revised to Signal-to-noise ratio (SNR) in Figure 1. The receiver structure has the inverse function of the transmitter structure. The 2 x 2 DM-based MIMO-FBMC received multimedia packets were demodulated using a 2 x 2 DM-based MIMO-FBMC demodulator, and the 2 x 2 DM-based MIMO-FBMC multimedia demodulation packets were extracted as output. The 2 x 2 DM-based MIMO-FBMC multimedia demodulation packets were the inputs to the adaptive BPSK/4OQAM demodulator, and the adaptive BPSK/4OQAM multimedia demodulation packets were extracted as output. The adaptive BPSK/4OQAM multimedia demodulation packets were the inputs to (2000,1000) LDPC decoders, and the voice, image, and data multimedia LDPC decoding signals were extracted as output.

[3] There is ‘2 UAMTA’ in line 149. I wonder what it means.

UAMTA has been revised to underwater acoustic multimedia communication architecture (UAMCA).

[4] There is a sentence in line 197 that the MIMO-FBMC channel estimation is assumed. I don't know what it means. Does it mean complete of channel estimation process?

The MIMO-FBMC channel estimation is assumed’ has been deleted. The MIMO-FBMC multipath path channel coefficients with channel estimation errors are inputted to the proposed simulation system, and the proposed system do not include channel estimation method.

[5] This paper considers 2x2 DM transmission mechanism, (2000, 1000) LDPC encoder, a power assignment mechanism, an OCPN mechanism, adaptively BPSK modulation and 4-OQAM methods. I think it is necessary to explain clearly what the new proposal is.

Please refer the abstract section.

[6] From the simulation results, the advantages of the proposed algorithm are not clear. If the channel estimation error rate is high, it is not surprising that the performance of the communication system is poor. Simulation results should be added to clearly show the advantages of the proposed technique.

Please refer the abstract section. In addition, the details performance of the proposed system has been demonstrated in the simulation section.

Round 2

Reviewer 2 Report

The autrhor has modified these problems I have mentioned. The paper feels better now.